# *TRAVEL*: Tag-Aware Conversational FAQ Retrieval via Reinforcement Learning

**Yue Chen♠,    Dingnan Jin♣,    Chen Huang♠,    Jia Liu♣,    Wenqiang Lei♠†**
♠ College of Computer Science, Sichuan University
♣ Ant Group, China
{chenyueeee24, huangc.scu, wenqianglei}@gmail.com
{dingnan.jdn, jianiu.lj}@antgroup.com

## Abstract

Efficiently retrieving FAQ questions that match users' intent is essential for online customer service. Existing methods aim to fully utilize the dynamic conversation context to enhance the semantic association between the user query and FAQ questions. However, the conversation context contains noise, e.g., users may click questions they don't like, leading to inaccurate semantics modeling. To tackle this, we introduce tags of FAQ questions, which can help us eliminate irrelevant information. We later integrate them into a reinforcement learning framework and minimize the negative impact of irrelevant information in the dynamic conversation context. We experimentally demonstrate our efficiency and effectiveness on conversational FAQ retrieval compared to other baselines.

## 1   Introduction

Retrieving FAQ questions that match users' intent during user-system conversations is critical for online customer service in large companies (e.g., Alibaba, Amazon). In this paper, we call this scenario **Conversational FAQ retrieval**. Normally, it employs an AI assistant to model user behaviors (e.g., queries and clicks) and iteratively retrieve FAQ questions until meeting users' intent (Ranoliya et al., 2017; Vishwanathan et al., 2023). To satisfy user experience (Gao et al., 2022), it is needed to design an efficient retrieval strategy to find FAQ questions that match user intent in minimal turns.

Current methods focus on modeling the semantic information in the conversation context. They model the semantic similarity to perform FAQ retrieval by concatenating user queries and clicking questions (Rosset et al., 2020; Vishwanathan et al., 2023) or applying attention mechanisms (Li et al., 2019). Although these methods show promising results, they assume that users' behaviors strictly adhere to their intent. In fact, users may click questions containing information that is irrelevant to their intent due to domain unfamiliarity or misoperation (Keyvan and Huang, 2022). This information, which we call "**tags**"[1] following (Yu et al., 2019; Romero et al., 2013), brings noise to the conversation context and disturbs the retrieval efficiency.

Taking Figure.1 (a) as an example, the user's target question is "How to cancel the automatic repayment of credit cards?". Due to domain unfamiliarity, in Figure.1 (c), the user clicks the question "How to cancel the automatic payment of bank cards", because it contains the same tag ("cancel") with the user's intent. Unfortunately, this introduces irrelevant tags such as "automatic payment" and "bank card". As a result, the system retrieves irrelevant FAQ questions in subsequent turns (e.g., "How to cancel the automatic payment of the credit cards?" in the next turn). Therefore, such irrelevant information makes existing systems require more turns to hit the FAQ question that matches the user's intent.

We believe that the key to finding the right FAQ question in minimal turns is to reduce the impact of irrelevant tags in clicked questions. Thus, we need to estimate whether a tag is irrelevant to user intent. Accordingly, as shown in Figure.1 (b), we assume that a tag has a high probability of being irrelevant if the user seldom clicks FAQ questions containing this tag. Besides, as shown in Figure.1 (e), the probability of a tag being irrelevant can be gradually estimated along with the dynamics of the conversation. This motivates us to utilize reinforcement learning (RL) to model the dynamic changes of the tags' irrelevance estimation. By maximizing the cumulative reward based on the estimation, the RL model learns an optimized strategy to reduce the impact of irrelevant tags and obtain the user intent in minimal turns. More specifically, we convert the conversation context into a representation

---

† Corresponding author.

[1]Tags take forms of keywords or segments in questions.

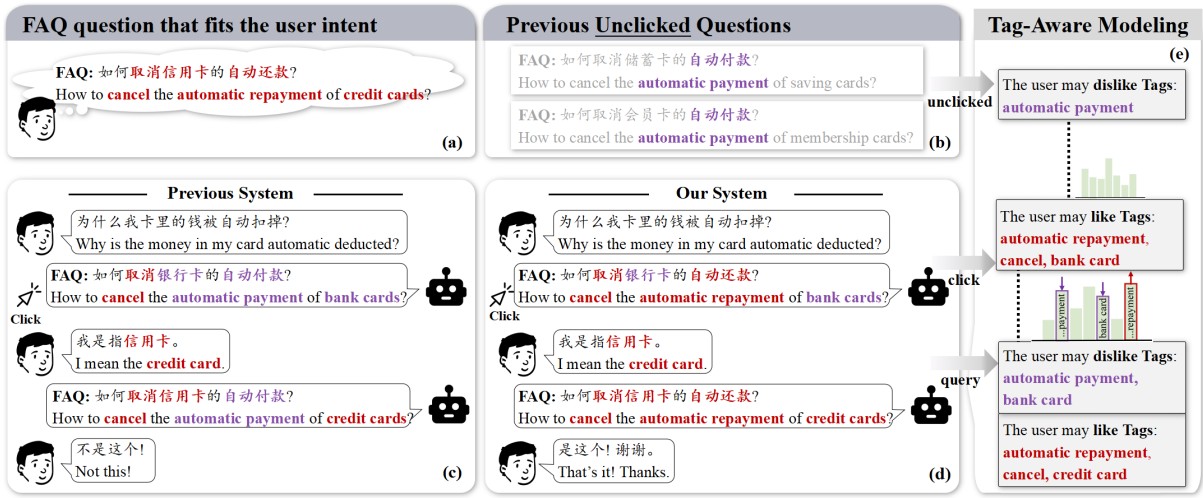

Figure 1: An illustration of conversation FAQ retrieval in the finance domain. The red and purple texts describe the relevant and irrelevant tags in the context, respectively. Figure (a) shows the user's desired FAQ question. Figure (b) shows that the user's past unclicked questions contain the tag automatic payment. In Figure (c), the user clicks a question containing automatic payment in the first turn, leading the system to retrieve irrelevant questions containing automatic payment in the next turn, resulting in user undissatisfaction. In Figure (e), our tag-aware modeling infers that the user doesn't like automatic payment according to the previous unclicked questions in Figure (b). As a result, in Figure (d), our system doesn't retrieve the irrelevant FAQ questions containing automatic payment.

that models the dynamic tags' irrelevance estimation. When taking the representation as the state, we **punish** the RL system when it retrieves FAQ questions containing irrelevant tags, and **reward** it when it retrieves users' desired FAQ questions in minimal turns. By doing so, the RL system can dynamically adjust its strategy to avoid retrieving questions that contain tags with high estimated irrelevance and achieve a successful retrieval as quickly as possible. In this way, we can effectively decrease the negative impact of irrelevant information in the conversation context, and retrieve users' desired FAQ questions in minimal turns. We call our method the Tag-aware conveRsational FAQ retrieVal via rEinforcement Learning (*TRAVEL*).

To sum up, we have the following contributions: (1) For the first time, we point out the significance of reducing the impact of irrelevant information introduced by noisy user behaviors in Conversational FAQ Retrieval. (2) We propose a tag-aware reinforcement learning strategy that models the dynamic changes of the tags' irrelevance to achieve successful FAQ retrieval in the minimal turn. (3) By developing new FAQ data, we test our method with intensive empirical studies. The results support the efficiency and effectiveness of our method.

## 2 Framework

**Notation.** We denote the collection of question-answer pairs as $FAQ = \{(q_1, a_1), ..., (q_n, a_n)\}$,

where $q_i$ is a FAQ question with $a_i$ as its answer. Each question $q_i$ is categorized into a set of tags $P_{q_i} = \{p_{i1}, p_{i2}, ...p_{im}\}$. For a user $u_i$, there is a question $q_i$ that matches his/her intent, which needs to be retrieved. At the turn $t$ of a conversation, the system performs retrieval based on the conversation context. The conversation context at the turn $t$ consists of: $H_t$, which records the user's queries; $Q_{click}^t$, which records the questions that user clicked; $Q_{rej}^t$, which records the questions that user ignore; The conversation ends if the system retrieves the question $q_i$. Otherwise, it continues to retrieve until reaching the maximum turns $T$.

**Framework Overview.** As depicted in Figure 2, We formulate *TRAVEL* as a multi-turn tag-aware reinforcement learning framework, which aims to learn a promising policy $\pi^* = \arg\max_{\pi \epsilon \Pi} \mathbb{E}\left[\sum_{t=0}^{T} r(s_t, \mathbf{a}_t)\right]$. Here, the action $\mathbf{a}_t$ indicates which FAQ question to retrieve from the candidates, and $s_t$ capture the conversation context at the turn $t$. With the help of RL, *TRAVEL* can learn an efficient strategy to perform successful retrieval in minimal turns. Our *TRAVEL* contains two key components, i.e., Tag-Level State Representation and Conversational Retrieval Strategy Optimization. The former estimates irrelevant tags in the context and converts the conversation context into the state; The latter optimizes a retrieval strategy by RL given the state. Overall, we begin by elaborating on the RL environment setting of

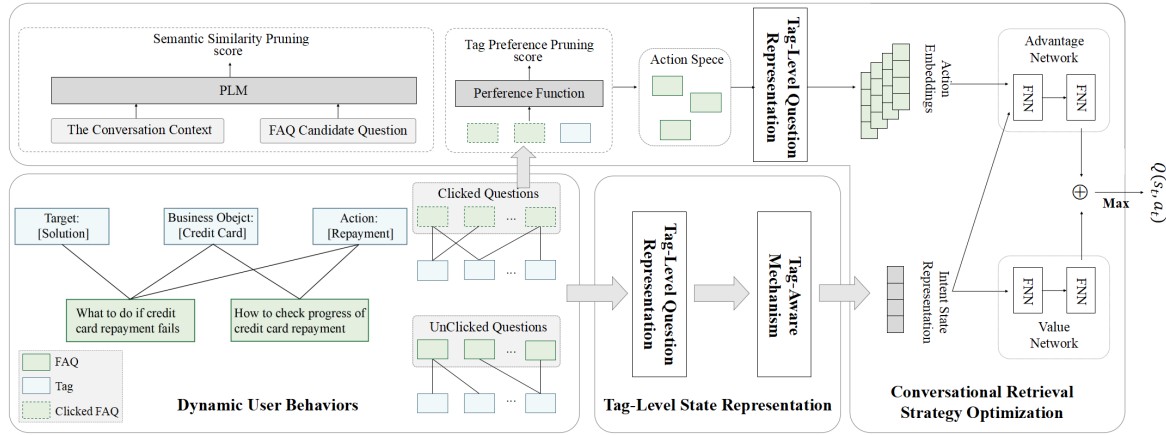

Figure 2: The overview of *TRAVEL*, for conversational FAQ retrieval.

*TRAVEL* in section 2.1. Then, in section 2.2, we delve into the two key components of *TRAVEL*.

## 2.1 RL Environment Setting

We elaborate on how to formulate the conversational FAQ task into RL. It involves informing the system about the state and actions (i.e., questions to retrieve), transitioning between states, and providing feedback-based rewards.

**State.** The state, formulated as $s_t = \{H^t, Q^t_{click}, P^t_{click}, Q^t_{rej}, P^t_{rej}\}$, containing the conversation context up to the turn $t$. Here, $Q^t_{click}$ is a set of user-clicked questions, and $P^t_{click}$ is a set of tags associated with questions in $Q^t_{click}$. Moreover, $Q^t_{rej}$ and $P^t_{rej}$ are the questions that the user did not click and their corresponding tags.

**Action.** Given the state, the system takes an action $\mathbf{a}_t$ by finding out which FAQ question should be retrieved from the candidate set $Q_{cand}$. In practice, we retrieve five questions at each turn.

**Transition.** When the user clicks/ignores questions or launches a query at turn $t$, our conversation context changes. Namely, the state $s_t$ is updated to a new state $s_{t+1}$, specifically, by adding the clicked $q_i$ and its tags to $Q^{t+1}_{click}$ and $P^{t+1}_{click}$, or appending them into the $Q^{t+1}_{rej}$ and $P^{t+1}_{rej}$ if the question is not clicked. If the user launches a query at turn $t$, we add the new query into conversation history $H^{t+1}$.

**Noise-Aware Reward.** To *achieve successful retrieval in minimal turns*, we reward the model when the retrieval succeeds and punish it when it fails or the turn number exceeds the maximum turns. It is also crucial to inform models when irrelevant tags are introduced by noisy user behavior, such as clicking questions with irrelevant tags. It enables models to adapt the strategy during conversations. In this paper, we propose five rewards: (1) $r_{click\_suc}$,

a positive reward when the user clicks. However, if clicked questions contain irrelevant tags, the value of this reward is reduced. (2) $r_{click\_fail}$, a negative reward when the user does not click any question. (3) $r_{ret\_suc}$, a strong positive reward when the user successfully obtains his target question, (4) $r_{extra\_turn}$, a negative reward when the number of turns increases, (5) $r_{quit}$, a strong negative reward when reaching the maximum turns.

## 2.2 Components of *TRAVEL*

*TRAVEL* consists of two components: Tag-Level State Representation and Conversational Retrieval Strategy Optimization. The Tag-Level State Representation component focuses on estimating irrelevant tags within the conversation context and transforming the context into the state representation. The Conversational Retrieval Strategy Optimization utilizes the state to determine a strategy for FAQ retrieval using Q-Learning, aiming to achieve accurate retrieval in minimal turns. It further enhances the RL process by pruning the action space, following previous work (Lei et al., 2020b).

### 2.2.1 Tag-Level State Representation

This section explains the estimation of irrelevant tags in the context and the modeling of the conversation context at the tag level to obtain the state representation. The process involves two steps. First, graph representations of questions at the tag level are obtained. This eliminates irrelevant semantics within questions and captures correlations between tags and questions. Then, a tag-aware mechanism is proposed to model the conversation context. It estimates irrelevant tags in the context and obtains the state in a fine-grained way. This state records the dynamically estimated irrelevance of tags.

**Tag-Level Question Representation.** To eliminate irrelevant information and utilize correlations between questions and tags, a graph $\mathcal{G}$ is constructed with questions and tags as nodes. The node representations of a question $q_i$ and a tag $p_i$ are obtained using TransE (Bordes et al., 2013), denoted as $\mathbf{e}_{q_i}$ and $\mathbf{e}_{p_i}$.

**Tag-Aware Mechanism.** We employ the tag-aware mechanism is employed to transform the conversation context into the state. This mechanism estimates whether a clicked question in the context contains irrelevant tags and calculates a weight to reflect this information. By doing so, the model becomes aware of the presence of irrelevant tags in the click question and implicitly eliminates them. The mechanism is defined as follows:

$$\mathbf{v}_n^t = \mathbf{W}_n * \mathbf{g}_n^t, \qquad (1)$$

$$\mathbf{g}_n^t = \frac{1}{|Q_{rej}^t|} \sum_{n \in Q_{rej}^t} \mathbf{e}_{q_n} + \frac{1}{|P_{rej}^t|} \sum_{n \in P_{rej}^t} \mathbf{e}_{p_n}, \qquad (2)$$

where $\mathbf{W}_n \in \mathbb{R}^{d \times d}$ is a trainable parameters. The $\mathbf{v}_n^t$ is derived from unclicked questions and their tags that are ignored by users. This information captures tags information that contradicts users' intent, referred to as *negative embedding*. Furthermore, given the graph embedding of the clicked questions $Q_{click}^t = [\mathbf{e}_{q_{i1}}, \mathbf{e}_{q_{i2}}...\mathbf{e}_{q_{in}}]$, we obtain the state representation $\mathbf{s}_t$ as follows:

$$\mathbf{s}_t = \sum_{n=1}^{\mathcal{N}} \alpha_n \mathbf{e}_{q_n}, \qquad (3)$$

$$\alpha_n = \frac{exp\left(\mathbf{h}^T \sigma\left(\mathbf{W}\left(\mathbf{v}_n^t || \mathbf{e}_{q_n}\right)\right)\right)}{\sum_{n'=1}^{\mathcal{N}} exp\left(\mathbf{h}^T \sigma\left(\mathbf{W}\left(\mathbf{v}_n^t || \mathbf{e}_{q_{n'}}\right)\right)\right)}, \qquad (4)$$

where $\mathbf{h}^T$ and $\mathbf{W}$ are trainable metrics.

The $\mathbf{s}_t$ is calculated by combining the clicked question embeddings $\mathbf{e}_{q_{in}}$ using weights. Each weight $\alpha_n$ is determined based on the score between $\mathbf{v}_n^t$ and clicked question embedding $\mathbf{e}_{q_{in}}$. A higher weight indicates the presence of more irrelevant tags in the question within the context.

It's important to note that although this weight is applied to questions, it is implicitly mapped onto the corresponding tags of questions through the graph representation. Using the weight contained in $\mathbf{s}_t$ as a signal, our model can implicitly eliminate irrelevant information in the context.

#### 2.2.2 Conversational Retrieval Strategy Optimization

Given the state $\mathbf{s}_t$ contains the information about the conversation context and irrelevant tags, we need a strategy to retrieve the right question in minimal turns. Thus, we utilize the Dueling Q-network following previous work (Zhou et al., 2020). The Dueling Q-network is formulated as:

$$Q(\mathbf{s}_t, \mathbf{a}_t) = f_{\theta_V}(\mathbf{s}_t) + f_{\theta_A}(\mathbf{s}_t, \mathbf{a}_t), \qquad (5)$$

where $f_{\theta_V}(.)$ and $f_{\theta_A}(.)$ are two separate multi-layer perceptions with parameters $\theta_V$ and $\theta_A$. This equation takes the state $\mathbf{s}_t$ and an action (the graph representation of a FAQ question) as inputs and provides a score for that question. The score $Q(\mathbf{s}_t, \mathbf{a}_t)$ represents the expected reward when taking the action $\mathbf{a}_t$ based on the state $\mathbf{s}_t$. To maximize the cumulative expected reward, we follow the Bellman equation (Bellman and Kalaba, 1957):

$$y_t = \mathbb{E}_{s_{t+1}}\left[r_t + \gamma \max_{\mathbf{a} \in \mathcal{A}} Q^*(\mathbf{s}_{t+1}, \mathbf{a}_{t+1})|\mathbf{s}_t, \mathbf{a}_t\right], \qquad (6)$$

where $y_t$ denotes the $Q^*(\mathbf{s}_t, \mathbf{a}_t)$. Since the reward is associated with retrieval accuracy, irrelevant tags information, and the number of turns as mentioned in section 2.1, maximizing the expected reward promotes the strategy to achieve accurate retrieval in minimal turns while avoiding retrieving other questions that contain irrelevant tags. Ultimately, for each action $\mathbf{a}_t$, which corresponds to an FAQ question, the system selects the FAQ question with the highest Q-value to retrieve.

**Two-Step Pruning Strategy.** The performance of reinforcement learning is compromised by a large action space (Lei et al., 2020a). Thus, we propose two pruning strategies to shrink the action space. (1) We utilize semantic similarity pruning to shrink the action space following baselines (Vishwanathan et al., 2023). It models the semantics association between queries and FAQ questions. We select top-$k_s$ FAQ questions with the highest semantic scores, forming the candidate set $Q_{cand\_sim}^t$. (2) Then, we use tag preference pruning to further reduce the action space. We choose top-$k_v$ questions with the highest score to form $Q_{cand\_pre}^t$ following the formula in Appendix A.5.

### 3 Experiments

This paper focuses on proposing a strategy to retrieve the appropriate FAQ question within a minimal number of turns. Consequently, our first research question investigates whether TRAVEL can outperform the baselines (including ChatGPT) given the limited turns. Subsequently, we explore

whether TRAVEL can maintain stable and superior performance in the face of different levels of noisy user behavior. Furthermore, we validate the efficiency of *TRAVEL*'s individual components, ensuring their impact on performance. Finally, we verified if the tag-aware mechanism can actually estimate whether and how many irrelevant tags a clicked question contains in the context. The above research questions are as follows:

- **RQ1:** Can TRAVEL outperform baselines in achieving more accurate retrieval given the limited number of turns?

- **RQ2:** How does our method perform in the presence of different levels of noisy user behaviors compared to baselines?

- **RQ3:** What is the impact of each component of TRAVEL on its performance?

- **RQ4:** Can the tag-aware mechanism effectively estimate the presence and quantity of irrelevant tags the clicked questions contain?

## 3.1 Dataset

We conduct experiments on our proposed data, since existing FAQ datasets do not contain the question tags and conversation history. Our dataset contains 72,013 conversation sessions. Each session is formulated as $(u_i, q_i, H_i)$, where $q_i$ represented the user's intended question, $H_i$ contains the conversation history between the user and system, and $u_i$ denoted the user along with their profile. There are 1449 FAQ questions and 1201 tags in the dataset. On average, each question contains 6 tags. See Appendix A.6 for more details of the dataset.

## 3.2 Experimental Settings

### 3.2.1 User Simulator

Due to the interactive nature, online experiments where the system interacts with real users and learns from their behaviors would be ideal. However, the trial-and-error strategy for training RL (Zhou et al., 2020) online would degrade the user's experience and the system's profit. Thus, following Lei et al. (2020c); Zhang et al. (2021), we simulate users' behaviors via a simulator.

Given a user $u_i$ and their target question $q_i$, we simulate their click behaviors. Users' clicking behavior is influenced by two main factors: their intent (target) (Zhang et al., 2021) and their profile (Zhou et al., 2020). Specifically, users are inclined

to click on items relevant to their intent or based on their interests (defined by their profiles). Therefore, the probability of user $u_i$ clicking a FAQ question $q_k$ is calculated as follows:

$$r(u_i, q_k) = \alpha * r_{ki} + (1 - \alpha) * c_{ki}, \quad (7)$$

$$c_{ki} = f(q_k, \mathbf{e}_i), \quad (8)$$

where: (1) $r_{ki}$ represents the probability of user $u_i$ clicking $q_k$ based on *relevance* to their target question $q_i$. We define relevance as the number of overlapping tags between $q_k$ and the user's target $q_i$. The click probabilities for different levels of relevance are derived from online statistics, as presented in Appendix A.4. (2) $c_{ki}$ represents the probability of user clicking $q_k$ based on their interest profile $\mathbf{e}_i$. It is modeled as $c_{ki} = f(q_k, \mathbf{e}_i)$. We train the function $f(.)$ using online data. Details of the hyperparameter $\alpha$ are in Appendix A.1.

### 3.2.2 Baseline

We compare the TRAVEL with two classes of baselines methods (comparisons with ChatGPT are in Appendix A.2). The first is FAQ retrieval which represents the standard way to retrieve FAQ questions. The second is Question Suggestion which is used in web searches to predict the next question users may ask, which has a similarity to FAQ retrieval in the form of the task.

**FAQ retrieval:** 1) BERT_TSUBAKI (Sakata et al., 2019)employs BERT to compute scores between queries and FAQ answers, and uses BM25 to compute scores between queries and FAQ questions; 2) SBERT_FAQ (Vishwanathan et al., 2023) is a fine-tuned BERT model optimized with triplet loss using FAQ questions; 3) DoQA (Campos et al., 2020) is a baseline for conversation-based question answering that utilizes only the first turn of the query; 4) CombSum (Mass et al., 2020) is a state-of-the-art FAQ retrieval method that calculates scores between the query and the question using both BM25 and BERT, and the score between the query and the answer using BERT; **Question Suggestion:** 5) CFAN (Li et al., 2019) is a multi-turn question suggestion method that takes queries and clicked questions as input; 6) KnowledgeSelect (Kim et al., 2020) is a multi-turn retrieval method in conversational settings that utilizes BERT to model the multi-turn user queries; 7) DeepSuggest (Keyvan and Huang, 2022; Rosset et al., 2020) is a standard question suggestion method for conversations that incorporates multi-turn user queries and clicked questions using BERT.

Table 1: The comparison among different conversation methods. Recall, SR and hDCG are the higher the better, while AT and AS are the lower the better. The comparison with ChatGPT is in the Appendix A.2.

| | Recall@5 | SR@2 | SR@3 | SR@4 | SR@5 | AT | AS | hNDCG |
|---|---|---|---|---|---|---|---|---|
| BERT_TSUBAKI | 0.3463 | 0.4461 | 0.5053 | 0.5496 | 0.5850 | 3.1526 | 15.7630 | 0.4304 |
| SBERT_FAQ | 0.5298 | 0.5569 | 0.5696 | 0.5824 | 0.5940 | 2.7513 | 13.7565 | 0.5167 |
| DoQA | 0.5212 | 0.5545 | 0.5796 | 0.6051 | 0.6254 | 2.7395 | 13.6977 | 0.5204 |
| CombSum | 0.5179 | 0.5788 | 0.6141 | 0.6449 | 0.6722 | 2.6443 | 13.2213 | 0.5515 |
| KnowledgeSelect | 0.4265 | 0.5252 | 0.5825 | 0.6274 | 0.6630 | 2.8385 | 14.1925 | 0.5043 |
| CFAN | 0.5213 | 0.5690 | 0.6145 | 0.6531 | 0.6864 | 2.6422 | 13.2109 | 0.5481 |
| DeepSuggest | 0.5398 | 0.6068 | 0.6711 | 0.7203 | 0.7592 | 2.4620 | 12.3101 | 0.5956 |
| TRAVEL | **0.5398** | **0.6179** | **0.6922** | **0.7586** | **0.8093** | **2.3915** | **11.9574** | **0.6177** |

### 3.2.3 Evaluation Metrics

Firstly, we evaluate the retrieval performance in the first turn using $Recall@5$. To assess the system's ability to successfully retrieve users' target questions within the limited $k$ turns, we utilize the metric *success rate (SR@k)* (Lei et al., 2020c). To measure the average number of turns the model takes in conversations, we use *AT(Average Turn)* (Lei et al., 2020c), and to evaluate the retrieval ranking performance, we employ *hNDCG@(T, K)* (Deng et al., 2021). Furthermore, considering the importance of user experience, we limit the exposure of users to a large number of questions, which could be burdensome. Hence, we introduce the Average Shown (AS) to quantify the average number of FAQ questions seen by users.

### 3.2.4 Implementation Details

We split the dataset by 4:1:1 for training, validation, and testing and set the number of retrieved questions at each turn as 5. We set the maximum turn $T$ as 10 during training. We set the $k_s$ of similarity-based as 100 and the $k_v$ of preference-based pruning as 10. We set the graph embedding size as 40. During the training procedure of DQN, the size of the experience buffer is 50000, and the sample size is 32. The learning rate is set to be 1e-4 with Adam optimizer. The discount factor $\gamma$ is set to be 0.99. We adopt the reward settings to train the proposed method: $r_{click\_suc} = 0.03 * m$, $r_{click\_fail} = -0.1$, $r_{ret\_suc} = 1$, $r_{extra\_turn} = -0.05$, $r_{quit} = -0.3$, where $m$ denotes the number of tags that coincide between the clicked question and the user's target question. When the user clicks a question that introduces noise, the $m$ is set to a minor value.

### 3.3 The overall comparison among different methods (RQ1)

This section presents the superior performance of TRAVEL compared to baselines in terms of achiev-

ing more accurate retrieval within limited turns. The results are presented in Table 1.[2] The result indicates that *TRAVEL outperforms all baselines.*

Analyzing Table 1 reveals that while FAQ retrieval methods can deliver satisfactory results in the initial turn ($Recall@5$), their performance noticeably declines in subsequent retrieval turns ($SR@k$) compared to TRAVEL. TRAVEL, on the other hand, outperforms these methods by achieving a 13.71% higher $SR@5$ and a 25.28% lower AT compared to CombSum. This highlights the limitations of existing FAQ retrieval methods in adequately capturing and utilizing the multi-turn conversation context. These methods require numerous turns for successful retrieval.

Furthermore, compared to existing question suggestion methods, TRAVEL also demonstrates superior performance. For instance, when compared to CFAN and DeepSuggest, which leverage multi-turn user queries and click questions, TRAVEL achieves 12.29%/5.01% higher $SR@5$ and 25.07%/7.05% lower $AT$, respectively. Additionally, TRAVEL outperforms these methods in terms of ranking effectiveness, with a 6.96%/2.21% higher $hNDCG$. These finds indicate that TRAVEL excels in ranking FAQ questions that align with the user's intent. The aforementioned experiments effectively demonstrate that TRAVEL outperforms state-of-the-art question suggestion methods by effectively utilizing the conversation context for FAQ retrieval.

### 3.4 The Noise Robust Testing (RQ2)

In this section, we evaluate the stable performance of TRAVEL in comparison to baselines across varying levels of noisy user behaviors. The results indicate that TRAVEL consistently maintains stable

---

[2]TRAVEL and DeepSuggest have consistent $Recall@5$. This is because there are no clicks in the first turn, so TRAVEL employs the same model as DeepSuggest. It can also use other models that have higher $recall$, which is not our focus.

Table 2: The ablation study. Recall, SR, and hDCG are the higher the better, while AT and AS are lower the better.

| | SR@2 | SR@3 | SR@4 | SR@5 | AT | AS | hNDCG |
|---|---|---|---|---|---|---|---|
| (a) - w/o semantic similarity pruning | 0.5802 | 0.6190 | 0.6555 | 0.6847 | 2.6054 | 13.0272 | 0.5589 |
| (b) - w/o tag preference pruning | 0.5966 | 0.6503 | 0.6931 | 0.7271 | 2.5202 | 12.6012 | 0.5798 |
| (c) - w/o tag-aware mechanism | 0.6113 | 0.6852 | 0.7525 | 0.8034 | 2.4113 | 12.0565 | 0.6140 |
| (d) - w/o tag-level question representation | 0.5565 | 0.5700 | 0.5826 | 0.5941 | 2.7511 | 13.7556 | 0.5167 |
| (e) - w/o noise-aware reward | 0.6149 | 0.6879 | 0.7555 | 0.8025 | 2.4018 | 12.0090 | 0.6140 |
| TRAVEL | **0.6179** | **0.6922** | **0.7586** | **0.8093** | **2.3915** | **11.9574** | **0.6177** |

and superior performance. To investigate this, we vary the weight $\alpha$ of the user simulator, setting it to 0.2, 0.5, 0.8, and 1. Appendix A.1 provides detailed experimental results that demonstrate how the user simulator deviates increasingly from real user behavior as the weights shift. This process can be considered as manually introducing noise. As illustrated in Figure 3, TRAVEL consistently achieves superior performance compared to the baselines, with an improvement of $4\%$ on SR@5, even when different levels of noise are introduced. In contrast, the performance of the baselines shows considerable fluctuations as the noise values change. Notably, DeepSuggest experiences a decrease of $5\%$ in SR@5 when the weight $\alpha$ transitions from 0.8 to 0.5. In summary, this experiment provides compelling evidence of TRAVEL's capability to effectively handle conversation contexts in the presence of varying levels of user-introduced noise.

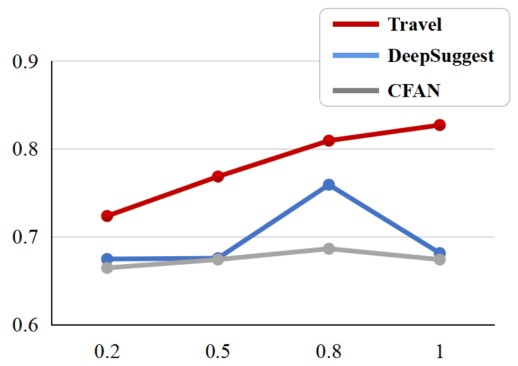

Figure 3: We observed the $SR@5$ (y-axis) for different $\alpha$ (x-axis). When $\alpha$ is 0.8, the noise is the lowest.

### 3.5 Ablation Study (RQ3)

#### 3.5.1 Tag-Level State Representation

We assess the effectiveness of modeling the conversation context at the tag level, specifically through the tag-level question representation and tag-aware mechanism. The experiments show the important role of these two components in the model performance, especially for the tag-level question representation. As shown in Table 2 row (d), removing

the tag-level question representation leads to a significant decrease in performance across all metrics. For instance, the $SR@5$ drops by $21.52\%$. This illustrates the importance of modeling the question representation at the tag level. Additionally, removing the tag-aware mechanism (Table 2 row (c)) results in a $5.9\%$ in $SR@5$. This highlights the benefit of incorporating the tag-aware mechanism. It is worth noting that the improvement brings by the tag-aware mechanism is smaller than that of the tag-level representation. This is because the tag-level representation has already captured the user's possible preferences for tags via the graph.

#### 3.5.2 Noise-Aware Reward

Next, we evaluate the effectiveness of the noise-aware reward $r_{click\_suc}$ (section 3.1). The experiment demonstrates that the reward setting enhances the performance of TRAVEL Specifically, we perform the ablation study by setting all clicked behaviors to receive the same reward. Table 2 row (e) illustrates that replacing the noise-aware reward leads to a $6.8\%$ decrease in $SR@5$. This demonstrates the significance of the noise-aware reward.

#### 3.5.3 Two-Step Pruning

The top part in Table 2 (rows(a-b)) presents the results when the proposed pruning strategies are omitted. Notably, all metrics experience a noticeable decline when discarding them. Specifically, without semantic similarity pruning, the SR@5 decreases by $12.46\%$. Similarly, omitting the tag preference pruning leads to an $8.22\%$ decrease in $SR@5$. These findings emphasize the critical role of utilizing semantic and tag information for effective pruning, which significantly enhances the performance of reinforcement learning (RL).

### 3.6 Validation of the Tag-Aware Attention Mechanism (RQ4)

In this section, we aim to verify the effectiveness of the tag-aware mechanism in estimating the presence and quantity of irrelevant tags in clicked ques-

tions. As explained in Section 2.2.1, the tag-aware attention mechanism is designed to assign a higher weight to inform the model when a clicked question contains many irrelevant tags.

To verify this, we compared the *weight value* of the Tag-Aware Mechanism in formula (3) with the weight value obtained from a fine-tuned sentence BERT. The sentence BERT takes the sentences of the clicked questions and the user's target question as input and outputs a score indicating the level of irrelevant information in the clicked questions. As depicted in Figure 4, the x-axis represents the degree of irrelevant tags in clicked questions of the context, where a larger value indicates fewer relevant tags and more irrelevant information. The comparison shows that tag-level modeling performs better in identifying the degree of irrelevant tags in the context. Specifically, as the degree increases from 0 to 10, the weight value obtained from the tag-aware mechanism exhibits a change of 24.12%. In contrast, the weight value obtained from the sentence BERT model only changes by 7.37%. This clearly demonstrates that our mechanism can better estimate the number of irrelevant tags in clicked questions compared to previous methods. These findings highlight the superiority of the tag-aware mechanism in accurately identifying irrelevant tags within the conversation context. This mechanism provides explanatory power for our approach and underscores its effectiveness.

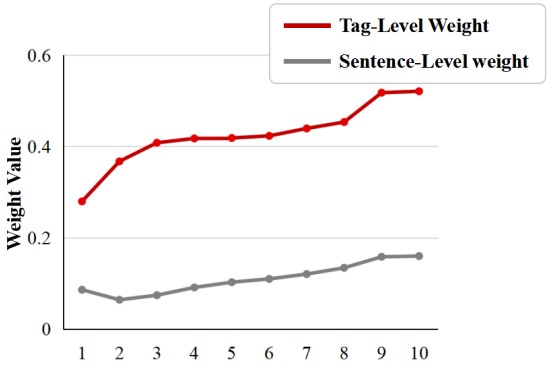

Figure 4: We observed the weight value (y-axis) for different levels of irrelevant information (x-axis).

## 4  Related Work

FAQ retrieval has broad applications in conversation such as conversational QA systems (Campos et al., 2020) and dialogue systems (Vishwanathan et al., 2023; Zhu et al., 2019). It is also widely used for online customer service in large companies such as Alibaba, Amazon, and Google. To satisfy

user experience in real-world applications, FAQ retrieval in conversation aims to perform successful retrieval in very few interactions. Technically, FAQ retrieval is achieved by modeling the semantic association between user queries and FAQ questions. Existing works use convolution neural networks (Karan and Šnajder, 2016), long short-term memory (Gupta and Carvalho, 2019) or pre-trained language models to model the semantic similarity (Sakata et al., 2019; Zhang et al., 2020). To fully utilize the conversation context, some web question suggestion methods (Li et al., 2019) incorporate the user-clicked questions with queries as the input of the semantic model using concatenation or attention. However, existing methods mainly model the conversation context from the semantic level, instead of modeling user behavior through a policy network. Meanwhile, existing datasets (Karan and Šnajder, 2016; Sakata et al., 2019; Zhang et al., 2020) do not contain tags or conversation history.

## 5  Conclusion

In this paper, we focus on proposing a strategy for retrieving FAQ questions that match user intent in minimal turns. To achieve this goal, we introduce a tag-aware conversational FAQ retrieval framework via reinforcement learning called *TRAVEL*. This framework is designed to eliminate the detrimental impact of irrelevant information in the conversation context. It contains two main components: Tag-Level State Representation and Conversational Retrieval Strategy Optimization. *TRAVEL* mitigates the negative impact of irrelevant information in the context by estimating its degree of irrelevance and employs a reinforcement learning strategy for performing FAQ retrieval. To verify our ideas, we create a dataset and develop a user simulator. Through extensive experiments, we justify the effectiveness of the *TRAVEL* framework, offering valuable insights into achieving successful FAQ retrieval in the fewest possible turns.

## 6  Limitation

Due to the interactive nature of our framework, online experiments where the system interacts with real users and learns from their behaviors would be ideal. However, the trial-and-error strategy for training RL online would degrade the user's experience and the system's profit. Therefore, we propose a user simulator to conduct the experiments offline. Even if we have proven the authenticity of the user

simulator, this still leaves us with a gap from the real scenario.

## 7 Ethics Statement

This paper presents a conversation FAQ retrieval framework with a new dataset and user simulator. Although our datasets are collected from an e-commerce company, they are designed for normal users and have been widely used by the public for some time. We also have carefully checked our dataset to make sure they don't contain any personally identifiable information or sensitive personally identifiable information (the user interest profile in Section 3.2.1 has been successfully desensitized). Thus, we believe there are no privacy concerns.

## Acknowledgement

This work was supported in part by the National Natural Science Foundation of China (No. 62272330); in part by the Fundamental Research Funds for the Central Universities (No. YJ202219).

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

## A  Appendices

### A.1  Evaluation of the User Simulator

Through experiments, we validate the authenticity of the user simulator and determine the value of $\alpha$ that best aligns with real user behaviors in formula (6). In this section, we assessed the fidelity of our simulator by comparing its behavior with actual user behaviors. We built a test set using online user click data, which included information on users and their clicked and unclicked FAQ questions. We then examined whether the simulator exhibited consistent click behavior with the users in this test set while varying the weight values ($\alpha$) in the formula

(6). As shown in Figure 5, we observed that the highest level of consistency between the simulator and the users in the test set was achieved when $\alpha$ was set to 0.8. Moreover, across different values of $\alpha$, the simulator consistently exhibited high levels of consistency (above 0.7). These experiments provide compelling evidence supporting the reliability of our simulator, indicating that it accurately emulates user behavior in the given context.

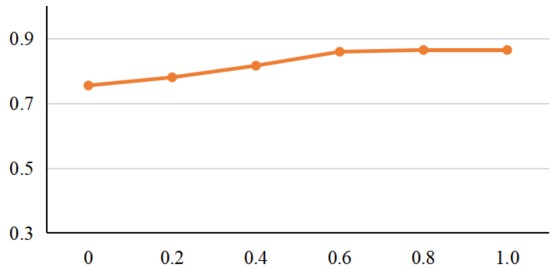

Figure 5: We observed the consistency (y-axis) between the simulator and the real user click behavior for different weights $\alpha$ (x-axis).

### A.2  The Comparison with ChatGPT

With ChatGPT's rapid development, we are curious about how well it would work in our scenarios. Due to the time-consuming inference of ChatGPT and the large length of the conversation, we did not test it on the complete testing set. We compare the performance of our method and ChatGPT on 1000 random samples from the complete testing set. It is worth noting that due to the limited input length of ChatGPT, we cannot feed all candidate questions as prompts to ChatGPT. Instead, only the 50 questions recalled by BM25 were used as input to ChatGPT. At the same time, we do an operation in favor of ChatGPT; that is, there must be the ground truth question in the recalled questions. This prevents ChatGPT from losing performance due to inaccurate BM25 recall. The experimental results are in Table 3. It shows that TRAVEL achieves better results (22.87% higher on $Recall@5$ and 12.83% higher on $SR@5$) compared to ChatGpt. This may be due to ChatGPT's lack of domain-specific a priori knowledge, which causes it to remain inadequate in domain-specific information retrieval.

### A.3  Parameter Sensitivity Analysis

The upper part of Table 4 summarizes the experimental results (SR) by varying the $k_s$, which denotes the size of the semantic similarity pruning. When the $k_s$ is 50, the performance essentially de-

Table 3: The performance comparison between the Chat-GPT and TRAVEL.

|           | Recall@5 | SR@5   | hNDCG  |
|-----------|----------|--------|--------|
| ChatGPT   | 0.3133   | 0.7207 | 0.4708 |
| TRAVEL    | **0.5420** | **0.8490** | **0.6482** |

Table 4: The effect of hyper-parameters.

|              | SR@2   | SR@3   | SR@4   | SR@5   |
|--------------|--------|--------|--------|--------|
| $k_s = 50$   | 0.5740 | 0.6204 | 0.6686 | 0.7178 |
| $k_s = 100$  | **0.6179** | **0.6922** | **0.7586** | **0.8093** |
| $k_s = 150$  | 0.6014 | 0.6685 | 0.7329 | 0.7832 |
| $k_v = 10$   | **0.6179** | **0.6922** | **0.7586** | **0.8093** |
| $k_v = 15$   | 0.6098 | 0.6813 | 0.7430 | 0.7931 |
| $k_v = 20$   | 0.6100 | 0.6762 | 0.7393 | 0.7820 |
| $T = 5$      | 0.6127 | 0.6880 | 0.7512 | 0.8025 |
| $T = 10$     | **0.6179** | **0.6922** | **0.7586** | 0.8093 |
| $T = 15$     | 0.6136 | 0.6919 | 0.7578 | **0.8106** |

creases. This may be because the similarity model is limited by ambiguous queries, resulting in some ground-truth FAQ questions being over-filtered. When the $k_s$ is 150, the performance also decreases. This is because the pruning might be ineffective when the pruning size is too large. The middle part of Table 4 summarizes the results by varying the $k_v$, which denotes the size of the preference-based pruning. The performance is better when we set the $k_v$ to a small value of 10. This result demonstrates the effectiveness of using tag preference pruning. The lower part shows the experimental results by varying the maximum turn $T$ during training. Although the maximum turn $T$ is set to 5 during testing, the performance is better when the $T$ is set to 10 during training. This is probably because the DQN can learn a better strategy when trained in a long conversation session. The performance decreases when the $T$ is set to a more significant value, such as 15. This may be because of the large gap in the conversation environment between training and testing when the length of the conversation session varies greatly.

## A.4 The Click Probabilities Table

This section shows the correlation between the probability of a user clicking and the tag overlap, which is calculated from online data mentioned in Appendix A.4. The results are shown in Table 5. It is important to note that this probability value only factors as part of the user simulator behavior.

Table 5: Corresponding click probabilities under different tag overlap degrees.

| Tag         | 0%      | 0%-20%   | 20%-40%   |
|-------------|---------|----------|-----------|
| Probability | 0.100   | 0.145    | 0.720     |
| Tag         | 40%-60% | 60%-80%  | 80%-100%  |
| Probability | 0.900   | 0.960    | 0.990     |

## A.5 Tag Preference Pruning

The score of tag preference pruning is calculated by $\sigma(s_i)$, where $s_i$ is determined by:

$$\sum_{q_n \in Q_{click}^t} e_{q_i}^T e_{q_n} + \sum_{p_n \in P_{click}^t} e_{q_i}^T e_{p_n} - \sum_{q_m \in Q_{rej}^t} e_{q_i}^T e_{q_m}, \quad (9)$$

Here, $e_{q_n}$ represents the graph embedding of the clicked question, $e_{p_n}$ represents the embedding of the corresponding tag of $e_{q_n}$, and $e_{q_m}$ represents the embedding of the question the user didn't click.

## A.6 Data Collection

Considering that existing FAQ datasets lack conversation history and question tags, to verify our studies, we propose a new dataset with the support of a large Chinese financial enterprise. We built this dataset in three steps. Firstly, we generated the FAQ questions. Next, we assigned tags to each FAQ question. Finally, we assembled conversation sessions, including user profiles and conversation histories, to create the training and testing sets. This table shows the statistics of our dataset. Specifically, the dataset has 1449 questions and 1201 tags. On average, there are 6 tags per question. The dataset contains 65,100 users and their conversation histories. In total, we collected 72,013 conversation histories. The average length of each conversation is four turns.

Table 6: Summary statistics of datasets.

| User  | Conversation | Question | Tag  |
|-------|--------------|----------|------|
| 65100 | 72013        | 1449     | 1201 |

### A.6.1 Questions Collection

We collect massive user questions and cluster them using the HDBSCAN algorithm (McInnes et al., 2017). Each cluster was then reviewed by domain experts who selected 10-15 representative questions to form the set of FAQ questions. The final collection consists of 1449 user questions, covering the majority of user intentions in the given domain.

### A.6.2 Tags Labeling

For tag labeling, we engaged three domain experts to pre-define the ontology of tags, which included business objects and user actions (e.g., business objects and user actions such as "Product" and "Intent" in Table 7). Using this ontology, we were able to assign multiple detailed tag values to each user question as shown in Table 7. To annotate the tag values, we initially assigned 20 crowd-workers to independently label ten randomly chosen questions. If the questions were simple or had few tags, another set of ten questions was provided. After the initial annotation, the workers resolved any disagreements and revised the annotation scheme. Subsequently, they collaboratively annotated an additional 30 questions, resulting in a high Fleiss kappa score of 0.691 (Fleiss and Cohen, 1973). Finally, these annotators proceeded to label the remaining questions, resulting in a high-quality corpus of 1449 questions with 1201 tag values.

Table 7: An example of labeling the tags of a question.

| What should I do if my credit card is overdue? | | |
|:---:|:---:|:---:|
| **Product** | **Movement** | **Intent** |
| Credit Card | Overdue | Solution |

### A.6.3 Conversation Collection

Once the constructed FAQ questions were online, we collect 72013 conversation session data for training and testing. Each session is formulated as $(u_i, q_i, H_i)$, where $q_i$ represented the user's intended question, $H_i$ contains the conversation history between the user and system, and $u_i$ denoted the user along with their profile. Specifically, to determine the target question $q_i$ for each user, we gathered the FAQ questions they had clicked on from the online data, which we formulated as $Q_{click} = \{q_1, q_2, ...q_n\}$. Among these clicked questions $Q_{click}$, we obtain $q_i$ based on the following rules: (a) The user terminated the conversation after clicking on $q_i$. (b) The user refrained from conducting further searches for a certain period of time following the click. (c) The user did not seek assistance from human customer service after the search session. Once we obtained $q_i$, we conducted a manual review to ensure its quality.