# OpenReview forum: "TRAVEL: Tag-Aware Conversational FAQ Retrieval via Reinforcement Learning"
_EMNLP/2023/Conference — EMNLP 2023 Main_

### Official Review · Reviewer_LGK8 · 2023-08-02

**Soundness:** 3

**Excitement:**

4: Strong: This paper deepens the understanding of some phenomenon or lowers the barriers to an existing research direction.

**Paper Topic And Main Contributions:**

This paper focuses on conversational FAQ retrieval, where the goal is to retrieve the appropriate FAQ question that matches the user's intent within a minimal number of conversation turns. The main contributions are:

1. Proposes a tag-aware conversational FAQ retrieval framework called TRAVEL that uses reinforcement learning to eliminate the negative impact of irrelevant information introduced through noisy user behaviors during the conversation.

2. Introduces two key components in TRAVEL: Tag-Level State Representation and Conversational Retrieval Strategy Optimization. The first estimates irrelevant tags in the context and converts it into a state representation. The second uses Q-Learning to determine an optimal retrieval strategy given the state.

3. Creates a new FAQ dataset with conversation histories and question tags to evaluate the framework.

4. Shows through experiments that TRAVEL outperforms baselines in achieving more accurate retrieval within limited turns. It also maintains superior performance across varying levels of noisy user behavior.

**Questions For The Authors:**

Q1) Could you discuss the prospects for extending this framework to more diverse FAQ datasets from different domains? What adaptations maybe needed?

Q2) Can you provide more analysis of how the tag relevance estimates change over the course of the conversational episodes? Some examples could be insightful.

Q3) Have you considered deploying this in a real interactive setting with humans for more rigorous evaluation? What would be the challenges?

**Reasons To Accept:**

1. Addresses an important real-world problem of improving conversational FAQ retrieval, which has many applications in conversational agents and customer service.

2. Proposes a novel tag-aware reinforcement learning framework as a solution, leveraging fine-grained tag information to deal with noisy behaviors. This is a technically sound approach.

3. Provides useful new insights on reducing the impact of irrelevant information during conversational retrieval through implicit modeling of tag relevance. Introduces a new FAQ dataset with conversation history and fine-grained tags, which could enable future research.

4. Achieves significantly improved results over baselines on the dataset across metrics like success rate and turns. This demonstrates the effectiveness of the framework.

**Reasons To Reject:**

1. The dataset is collected from a single commercial domain, which may limit the generalizability of the results. Testing on more diverse FAQ data could further validate the approach.

2. There could be more analysis on the learned strategy to provide better understanding, e.g. how tag relevance estimates change over time.



**Reproducibility:**

3: Could reproduce the results with some difficulty. The settings of parameters are underspecified or subjectively determined; the training/evaluation data are not widely available.

**Reviewer Confidence:**

3: Pretty sure, but there's a chance I missed something. Although I have a good feel for this area in general, I did not carefully check the paper's details, e.g., the math, experimental design, or novelty.

**Typos Grammar Style And Presentation Improvements:**

Line 96: Missing space before TRAVEL


Line 115: FAQ should be FAQs

Line 270: Missing period at the end of sentence

Line 515: it should be its

The writing is clear overall and the paper is well organized. The background and related work provide good context. More intuition could be provided when introducing the technical details.

---

> ### Author Rebuttal · Authors · 2023-08-29
>
> Thank you for your positive review. We will address your questions point by point.
>
> $\text{\color{blue} Q1: Could you discuss ... extending this framework to FAQ datasets from different domains? What adaptations may be needed?}$
>
> **Answer:** It is not difficult to extend TRAVEL to different domains. These adaptations are at the data level and revolve around the construction of domain-specific datasets. This process involves collecting conversational data and annotating tags for FAQ questions. Obtaining the former data can be achieved through collaboration with various companies or online services, whereas the latter task can be efficiently completed using automated tools for keyword extraction such as [1], or human annotation as we did.
>
> $\text{\color{blue} Q2: ... provide analysis of how the tag relevance estimates change over ... conversational episodes?}$
>
> **Answer:** Thank you for your insightful suggestions! As suggested, we sampled 5000 conversational data. For each data, we randomly selected a tag that is irrelevant to the user intent and a relevant tag. We then analyze how the relevance score of these two tags changes over the conversational episode, as shown in the table below.
>
> Turn_count                               | 1    | 2    | 3    | 4    | 5    |
> | ---------------------------------------- | ---- | ---- | ---- | ---- | ---- |
> | Relevance Score for a **irrelevant** tag | 0.51 | 0.50 | 0.49 | 0.49 | 0.48 |
> | Relevance Score for a **relevant** tag   | 0.56 | 0.57 | 0.57 | 0.62 | 0.62 |
>
> These results also support our statement in line 66 that "the probability of a tag being irrelevant or not can be gradually estimated along with the dynamics of the conversation."
>
> More analysis will be added in the revised version.
>
> $\text{\color{blue} Q3: ... deploying this in a real-interactive setting ... What would be the challenge?}$
>
> **Answer:** A potential challenge is the Concept Drift[2], which refers to the phenomenon where the statistical properties of a target variable change over time. It makes a previously trained model less accurate or even obsolete in the real-interactive setting. This is a common challenge in the conversational information retrieval (CIR) community. Our future work will try to bridge the gap between offline and online environments by building more realistic interactions using LLMs.
>
> **Reference:**
>
> [1] Matej Martinc, Blaž Škrlj, and Senja Pollak. 2022.Tnt-kid: Transformer-based neural tagger for keyword identification. Natural Language Engineering, 28(4):409–448.
>
> [2] Lu, J., Liu, A., Dong, F., Gu, F., Gama, J., \& Zhang, G. (2018). Learning under concept drift: A review. IEEE transactions on knowledge and data engineering, 31(12), 2346-2363.

---

### Official Review · Reviewer_TqE8 · 2023-08-04

**Soundness:** 3

**Excitement:**

3: Ambivalent: It has merits (e.g., it reports state-of-the-art results, the idea is nice), but there are key weaknesses (e.g., it describes incremental work), and it can significantly benefit from another round of revision. However, I won't object to accepting it if my co-reviewers champion it.

**Paper Topic And Main Contributions:**

This paper proposed a tag-aware conversational FAQ retrieval method. It proposed to use tags of FAQ questions to eliminate irrelevant semantics.

**Reasons To Accept:**

The topic of this paper is very interesting and promising. They want to better align the semantics between queries and retrieved answers.

**Reasons To Reject:**

1. The main issue this paper wants to address is the utilization of conversation data when retrieved answers fail to satisfy the user's query. They introduce a tag for each retrieved answer. But the connection between the problem they want to solve and the solution is not very clear. Why introducing a tag can better align the user's query to retrieved answers? The motivation for the tag is not very well explained. The paper is written in a way that the tag is the only extra information they have, they use it and it performs better.

2. The method is only tested on one dataset, which is not clear about its generality.

**Reproducibility:**

2: Would be hard pressed to reproduce the results. The contribution depends on data that are simply not available outside the author's institution or consortium; not enough details are provided.

**Reviewer Confidence:**

4: Quite sure. I tried to check the important points carefully. It's unlikely, though conceivable, that I missed something that should affect my ratings.

**Typos Grammar Style And Presentation Improvements:**

It would be better to provide more comprehensive descriptions of tables and figures directly in their captions.

---

> ### Author Rebuttal · Authors · 2023-08-28
>
> Thank you for your detailed feedback and suggestions. We will address your concerns point by point.
>
> $\text{\color{blue} 1. Why introduce a tag ... The motivation for the tag is not well explained ...}$
>
> **Answer:** **(1)** Tags help estimate the irrelevant information in clicked questions, and **(2)** help identify whether a FAQ question matches the user intent. Related discussion can be found in lines 61-66.
>
> More specifically, it is crucial to have an efficient strategy to find FAQ questions that match user intent in minimal turns, especially when user behaviors could be noisy (Line 40-55). The noisy behaviors may involve irrelevant information that disrupts the user intent modeling. The irrelevant information often takes the form of keywords in clicked questions, which we refer them as **"tags"** following (Yu et al., 2019; Romero et al., 2013). For example, due to domain unfamiliarity (Keyvan and Huang, 2022), a user might click the question "How to freeze the credit card" while their actual intent is "How to close the credit card". This mistakenly clicked question contains the irrelevant tag "freeze", which could **disrupt the subsequent retrieval**, as shown in line 50. Besides, we found that in **47\%** of the conversations, users clicked questions that contained irrelevant tags. Consequently, there arises a need to estimate the relevance of tags within clicked questions and reduce the impact of those irrelevant tags.
>
> We statistically analyzed the connection between online user click patterns and tags, as shown in the table below. We first counted the number of each tag contained in the user click question. Then, we calculated the probability of tags being relevant to user intent under different numbers. As shown in the Table, **the probability of the tag being relevant is directly proportional to the number of clicked questions that contain this tag.**
>
> | The Number of clicked questions contain this tag      | 1    | 2    | 3    | 4    | 5    |
> | --------------------- | ---- | ---- | ---- | ---- | ---- |
> | Relevance probability of this tag | 0.55 | 0.67 | 0.75 | 0.80 | 0.85 |
>
>
> This observation grants us two advantages in utilizing tags:
>
> **(1) By combining tags and user behaviors, we can better estimate the irrelevant information in clicked questions.** In particular, if a user's historical clicked questions rarely contain a specific tag, this tag can be regard as irrelevant information in the current clicked question.
>
> **(2) Tags help identify whether a FAQ question matches the user intent.** A user who frequently clicks questions containing a specific tag is likely to have that tag relevant to his/her intent. Consequently, other questions that contain that tag could also match the user intent.
>
> Building upon this observation, we propose the tag-aware RL framework to model the dynamic changes of the tags' irrelevance estimation. We also experimentally demonstrated the effectiveness of our framework.
>
> $\text{\color{blue} 2. The method ... tested on one dataset ...not clear about its generality.}$
>
> **Answer:** We focus on the scenario of conversational FAQ retrieval, i.e., retrieving FAQ questions that match users’ intent during user-system conversations. This is prevalent in online services like XiaoAi, Siri, and Alexa [2,3,4].
>
> After investigation and research, we noticed that existing FAQ datasets lack tag and conversation histories (Karan and Šnajder, 2016; Sakata et al., 2019; Zhang et al., 2020), which makes them unavailable for conversation FAQ retrieval research.
> Hence, we took the first step to annotate a conversational FAQ dataset derived from the financial domain, as detailed in Appendices A.6. Notably, the task of data annotation is widely recognized as labor-intensive [1], which is why we had to test our method on one dataset. It's worth noting that our method was not specifically for the financial domain. It could be adapted to other domain datasets if available.
>
> In addition, we have described the details of our method (cf.Section 2.2) and the dataset construction (cf. Appendices). Thus, our method is clear and reproducible. We will also provide more comprehensive descriptions of tables and figures directly in our captions.
>
> **Reference:**
>
>
> [1] Alexander Sorokin and David Forsyth. 2008. Utility data annotation with amazon mechanical turk. In 2008 IEEE computer society conference on computervision and pattern recognition workshops, pages 1–8.IEEE.
>
> [2] https://xiaoai.mi.com/
>
> [3] www.siri.com/
>
> [4] https://www.alexa.com/

---

### Official Review · Reviewer_M6qM · 2023-08-11

**Soundness:** 3

**Excitement:**

3: Ambivalent: It has merits (e.g., it reports state-of-the-art results, the idea is nice), but there are key weaknesses (e.g., it describes incremental work), and it can significantly benefit from another round of revision. However, I won't object to accepting it if my co-reviewers champion it.

**Paper Topic And Main Contributions:**

The paper presents TRAVEL, a reinforcement learning based approach to retrieve FAQs most relevant to current conversational context of user.
Authors systematically study the impact of each component of TRAVEL.

**Reasons To Accept:**

This paper proposes a novel application of reinforcement learning framework in the relevant FAQ retrieval.

Authors carefully design state action and reward space.
Authors also conduct thorough evaluation of the proposed system.

**Reasons To Reject:**

Paper is an application of reinforcement learning framework to the problem of FAQ retrieval.

**Reproducibility:**

4: Could mostly reproduce the results, but there may be some variation because of sample variance or minor variations in their interpretation of the protocol or method.

**Reviewer Confidence:**

3: Pretty sure, but there's a chance I missed something. Although I have a good feel for this area in general, I did not carefully check the paper's details, e.g., the math, experimental design, or novelty.

---

> ### Author Rebuttal · Authors · 2023-08-28
>
> Thank you for your feedback and suggestions. We will address your concerns point by point.
>
> $\text{\color{blue} 1. Paper is an application of reinforcement learning ... not strong on novelty aspects ....}$
>
> **Answer:** We suppose that the reviewer may have misunderstood our contribution and novelty. We want to emphasize that our main contribution is to minimize the impact of irrelevant information from noisy user behaviors, thereby enhancing retrieval efficiency, as explained in line 104. This is a crucial but poorly addressed challenge in the current FAQ retrieval community. RL is a suitable method to help us address an issue in this challenge and can be replaced by other frameworks.
>
> $\text{\color{blue} 2. This paper ... should conduct evaluation ... in a real-time application.}$
>
> **Answer:** We follow the existing studies that employ a user simulator for evaluation, which is a well-established paradigm (Lei et al. (2020a); Zhang et al. (2021)).
> It has good reproducibility and enables us to have fair comparisons with existing methods that use simulators.
> We will try to conduct a real-time application in future work.

---

### Meta-Review · Area_Chair_RvJL · 2023-09-21

**Recommendation:** 4

**Metareview:**

The paper presents TRAVEL, a reinforcement learning-based approach to retrieve FAQs most relevant to the current conversational context of a user.

Overall, reviewers are satisfied with the contributions. The paper can be improved by adding results on more datasets, currently, only one dataset is experimented with.

---

### Decision · Program_Chairs · 2023-10-07

**Decision:**

Accept-Main

**Comment:**

The paper presents TRAVEL, a reinforcement learning-based approach to retrieve FAQs most relevant to the current conversational context of a user.

Overall, reviewers are satisfied with the contributions. The paper can be improved by adding results on more datasets, currently, only one dataset is experimented with.